# Life after Cleavage: The Story of a β-Retroviral (MMTV) Signal Peptide—From Murine Lymphoma to Human Breast Cancer

**DOI:** 10.3390/v14112435

**Published:** 2022-11-02

**Authors:** Jacob Hochman, Ori Braitbard

**Affiliations:** 1Department of Cell and Developmental Biology, Alexander Silberman Institute of Life Science, The Hebrew University of Jerusalem, Jerusalem 9190401, Israel; 2Department of Bioinformatics, The Faculty of Life and Health Sciences, Jerusalem College of Technology, Jerusalem 9372115, Israel

**Keywords:** breast cancer, MMTV, betaretrovirus, signal peptide, p14, immune therapy

## Abstract

An increasing body of evidence in recent years supports an association of the betaretrovirus mouse mammary tumor virus (MMTV) with human breast cancer. This is an issue that still raises heated controversy. We have come to address this association using the signal peptide p14 of the MMTV envelope precursor protein as a key element of our strategy. In addition to its signal peptide function, p14 has some significant post endoplasmic reticulum (ER)-targeting characteristics: (1) it localizes to nucleoli where it binds key proteins (RPL5 and B23) involved (among other activities) in the regulation of nucleolar stress response, ribosome biogenesis and p53 stabilization; (2) p14 is a nuclear export factor; (3) it is expressed on the cell surface of infected cells, and as such, is amenable to, and successfully used, in preventive vaccination against experimental tumors that harbor MMTV; (4) the growth of such tumors is impaired in vivo using a combination of monoclonal anti-p14 antibodies or adoptive T-cell transfer treatments; (5) p14 is a phospho-protein endogenously phosphorylated by two different serine kinases. The phosphorylation status of the two sites determines whether p14 will function in an oncogenic or tumor-suppressing capacity; (6) transcriptional activation of genes (RPL5, ErbB4) correlates with the oncogenic potential of MMTV; (7) finally, polyclonal anti-p14 antibodies have been applied in immune histochemistry analyses of breast cancer cases using formalin fixed paraffin-embedded sections, supporting the associations of MMTV with the disease. Taken together, the above findings constitute a road map towards the diagnosis and possible prevention and treatment of MMTV-associated breast cancer.

## 1. MMTV and Cancer

It is estimated that 15–20% of all cancers are associated with viruses. Viruses known to cause cancer in humans include the Epstein-Barr virus (EBV), hepatitis B virus (HBV), human T-cell lymphotropic virus type 1 (HTLV-1), human papilloma virus (HPV), hepatitis C virus (HCV), Kaposi’s sarcoma-associated herpes virus (KSHV) and the Merkel cell polyomavirus (MCV) [1]. The increasing interest and better understanding of the role of viruses in cancer pathogenesis led to the development of preventive vaccines against HBV and HPV, as well as directly contributed to fundamental discoveries in cancer biology.

Mouse mammary tumor virus (MMTV) is a member of the genus Betaretroviruses.

It causes mammary carcinoma (breast cancer) and lymphoma in mice [2,3,4,5].

Since the advent of PCR technology, an increasing body of evidence has supported the involvement of a virus that is genetically indistinguishable from MMTV (also named HMTV for human mammary tumor virus or HBRV for human Betaretrovirus) anywhere from 16–38% in human sporadic breast cancer [6,7,8,9]. However, this is a highly controversial area that we would like to address by using evidence based on the association of the p14 signal peptide of MMTV with human breast cancer.

### 1.1. MMTV Genome

The MMTV genome is organized into gag, pro, and pol genes, which overlap in different reading frames, and the Env gene [10]. The env gene is translated from a spliced mRNA, whereas the gag, pro, and pol genes are all translated from the same genome-size mRNA and ribosomal frameshifts are required for synthesis of the gag–pro and gag–pro–pol precursor polyproteins [10,11,12]. The gag and/or env gene products of the virus have been linked to transformation in mammary tumor-susceptible mice [13].

### 1.2. MMTV Infection and Life Cycle

MMTV is transmitted exogenously from infected mothers to suckling pups through milk. The viral particles reach the intestinal epithelium where they are trancytosed by M cells to dendritic cells and subsequently spread to T and B lymphocytes in Peyer’s patches of the gut [14,15,16,17], eventually leading to virus amplification and mammary gland infection. For additional details see the review by Parisi et al. in the current issue of Viruses [18].

Initially, the virus interacts with a cellular receptor protein such as transferrin receptor 1 (TfR1) expressed on the surface of infected mice cells [19,20]. TfR1 is highly expressed on activated lymphocytes and dividing mammary epithelial cells in vivo, which probably limits MMTV infection to these cell types [21,22,23]. Following interaction with its receptor at the cell surface, the virus is actually endocytosed along with the TfR1 to a late endosomal acidic compartment where the actual fusion of the virus takes place, not on the cell surface like other retroviruses. It is at this point that the capsid is released into the cytoplasm for further uncoating [24,25,26,27] (reviewed in Ross, 2010). The capsid contains the viral genomic RNA and other viral and cellular proteins (such as reverse transcriptase) that help convert the viral RNA into DNA. After uncoating, the viral genome is reverse-transcribed, is transported to the nucleus, and the provirus integrates into the genome. In the next step, the integrated viral DNA (proviral DNA) makes viral mRNAs and proteins that are used to make new viral particles in the cytoplasm. The Env membrane proteins are synthesized in the rough endoplasmic reticulum (RER) and trafficked through the Golgi network (GN). The immature virus particles then bud out of the cell. This is followed by maturation of the virus particles, making them infectious. While little is known about where and how MMTV virion assembly takes place, there is a lot more clarity on how the MMTV genomic RNA gets packaged into the virus particle using structural RNA elements on the genomic RNA [19,24,25,26,27].

### 1.3. MMTV in Breast Cancer

Cells isolated from ascites or pleural effusion of patients with metastatic breast cancer contained MMTV sequences in their DNA, expressed the MMTV Env protein and showed, by electron microscopy, retroviral particles similar to the mouse virus [28]. The same group reported the detection of HMTV proteins in human breast cancer cells that were 90–98% homologous to MMTV [29]. Moreover, MMTV-like genomic sequences have been detected in tumor samples from patients with lymphoma as well as in patients that contracted both diseases [30,31].

It is noteworthy that MMTV can infect human breast cells and propagate in vitro [32,33,34]. Additionally, saliva has been proposed as one route of inter-human infection by the virus [35]. Taken together, in all likelihood, MMTV seems to be involved in a significant percentage of human breast cancers with emphasis on geographical distribution [36,37]. It was proposed earlier and revisited lately that differences in geographical distribution of MMTV-associated breast cancer were linked to the distribution of particular strains of mice, supporting zoonosis [38,39].

However, there is an accumulation of reports that question the involvement of MMTV in human breast cancer. Some of these are presented and discussed in the following references [18,40,41,42] and additional manuscripts in the current issue of Viruses.

Independent of whether the linkage between MMTV and breast cancer is of a causative nature or representing an epiphenomenon, breast cancer constitutes a leading cause of cancer-related deaths in women worldwide. Indeed, according to the World Health Organization (WHO) the most common cancer in 2020 (in terms of new cases) was breast cancer, exceeding 2.25 million cases worldwide, resulting in a yearly mortality of 685,000 women.

The American Cancer Society’s estimates for breast cancer in the United States for 2021 were the following: about 281,550 new cases of invasive breast cancer and 49,290 new cases of ductal carcinoma in situ (DCIS). About 43,600 women were expected to succumb to the disease. It is noteworthy, that independent of breast cancer, HBRV was also reported in cases of the autoimmune liver disease primary biliary cholangitis [43,44,45].

An extended historical overview of the viral origin of human breast cancer (and primary biliary cholangitis) is given in the present issue of “Viruses” by Generoso Bevilacqua, one of the pioneers of the field [46].

## 2. The Lymphoma Connection

Our interest in the association of MMTV with murine lymphoma, mammary carcinoma and subsequently with human breast cancer, stemmed, through serendipity, from our earlier studies on the growth regulation of murine lymphoma and efforts to develop an experimental model for lymphoma xenogenization. Xenogenization is a term coined by Hiroshi Kobayashi to describe all attempts at making tumor cells antigenically foreign to their host, such that they might be used more effectively in immune therapy and diagnosis [47]. The experimental model we developed [48,49] consists of highly tumorigenic mouse lymphoma cells (named T-25) with a median survival of 17–20 days post intraperitoneal (IP) inoculation. These cells were derived from the S49 T-cell lymphoma (that harbor MMTV) and grown in suspension culture as single cells. From T-25 cells, we derived a substrate-adherent variant named T-25-Adh (Figure 1). These cells were non-tumorigenic in syngeneic BALB/c hosts. Furthermore, a single IP inoculation of live T-25-Adh cells into BALB/c mice immunized the recipients for practically a lifetime (tested up to 18 months) against a challenge with parental T-25 cells. We have shown earlier that the immunogenic potential of T-25-Adh cells was governed by differential morphogenesis of MMTV precursor cytoplasmic A particles in response to a challenge by αβ-interferon [50,51].

## 3. Enter the Signal Peptide, p14

In an effort to identify putative cell surface tumor associated antigens, live T-25-Adh cells were used to generate monoclonal antibodies (mAbs) [52]. Upon subsequent cell permeabilization, using immune fluorescence analysis, one of these mAbs, M-66, interacted strongly with nucleoli from both T-25 and T-25-Adh cells. Analysis of a cDNA expression library followed by protein microsequencing, identified the nucleolar (and cell surface) antigen as the 98 amino acid signal peptide of the envelope precursor protein of MMTV we named MMTV-p14 (p14 for short), in accordance with its electrophoretic mobility. This was the point in time that got us interested in this particular viral signal peptide. Following our initial findings [53,54,55] it was demonstrated that p14 functions as a nuclear export factor for intron-containing viral transcripts, similar to the properties of the HIV-Rev protein [56,57]. The envelope protein of MMTV is synthesized as a 73-kDa Env precursor. Its signal peptide (p14) is cleaved and the protein is further processed to give the envelope glycoproteins gp52 and gp36 (Figure 2). At first, it was proposed that a stretch of 19 amino acids immediately adjacent to the amino terminus of gp52, (comprising the current H&C domains-see below) could act as a signal peptide [58].

Following our reports on the nucleolar localization of p14, a 33 kDa spliced variant of the env mRNA was identified, the translation product of which was named regulator of export/expression of MMTV mRNA—Rem [56,57]. Rem shares identity with the envelope precursor protein in 98 signal peptide amino acids, as well as with the N-terminal (162 amino acids) and the C-terminal (41 amino acid residues) of Env, all in all 301 amino acids (Figure 2).

A set of studies by the group of J. Dudley [59,60,61,62,63] and Dultz et al. [64] demonstrated that Rem is directed to the ER membrane where the 98 aa sp–Rem (p14 in our terminology) is cleaved by signal peptidase followed by retrotranslocation (membrane extraction) to the cytosol with the participation of VCP/p97 ATPase. It then localizes to the nucleus/nucleolus (avoiding ubiquitination and proteasomal degradation) where it functions as a nuclear export factor for intron containing transcripts [56,57].

The translation product of another alternatively spliced variant is a p21 protein containing p14, plus a 7 kDa extension into gp52 (Figure 2), as it is recognized by both anti-gp52 and anti-p14 antibodies. However, this extension is not a direct continuation, since polyclonal antibodies raised against the direct continuation of the p14 sequence into gp52 did not recognize p21. In addition, p21 and p14 do not share a precursor/product relationship, based on pulse-chase experiments whereby both demonstrated the same kinetics of synthesis and degradation [55]. Up to now, p21 has been identified only in the highly tumorigenic T-25 lymphoma cells and in their direct descendants derived through serial in vivo transplantation in syngeneic hosts. Its significance is not known at present.

## 4. A Word on Signal Peptides

Signal peptides (usually around 16–30 amino acids long) mediate the targeting and insertion of nascent secretory and membrane proteins into the endoplasmic reticulum (ER). After fulfilling their ER targeting function, they are cleaved at their C-terminus by a signal peptidase and usually further degraded by signal peptide peptidases. Following the seminal studies of G. Blobel and colleagues [65,66,67], it was demonstrated that signal peptides are not just “greasy proteins” [68], as a growing number of viral signal peptides have been shown to carry out additional (post-ER targeting) functions, supporting the concept of bioactive viral SPs as modulators of their host cell physiology. For example, the signal peptides of several arenaviral glycoproteins (Lassa, Junin, and lymphocytic choriomeningitis virus) remain membrane-inserted. They are necessary for the processing of the mature glycoprotein complexes, and important for viral infection. In hepatitis C virus poly-proteins, signal peptide peptidase processing results in the release of the core protein into the cytosol and is essential for HCV assembly [69,70,71,72,73,74,75,76,77]. This also includes modulation of the host immune response, as in the case of human cytomegalovirus (HCMV), where the signal peptide (through regulation of cell surface expression of two NK cell ligands) differentially affected two distinct NK cell evasion pathways [78]. In another report [79], an intact signal peptide of dengue virus E protein (ER-targeted E protein) enhanced the immunogenicity for CD8+ T cells and generated superior antibody responses when expressed from the immunogenic recombinant vaccine vector, Vaccinia Ankara.

Alignment of the p14 sequence with the homologous sequences from mouse and human samples reveals an identity over 95% (Figure 3).

1. AAC16282.1 (Mouse mammary tumor virus), 2. P03374.1 (Mouse mammary tumor virus (STRAIN GR)), 3. ABB02515.1 (Mouse mammary tumor virus), 4. GenBank: QDS02904.1 (Cloning vector pHYB-TBLV), 5. AFZ15783.1 (human Betaretrovirus), 6. AAP73834.1 Env, partial (human Betaretrovirus) 7. AAP73829.1 Env, partial (human Betaretrovirus), 8. P10259, 9. Q85646, 10. P03374

## 5. p14 Is Not an Orphan Nucleolar Signal Peptide

MMTV is a member of the genus Betaretroviruses. Two other members of the genus are the human endogenous retrovirus-K (human mouse mammary tumor virus-like-2) (HERV-K (HML-2)) [80] associated with germ cell tumors and melanoma, and the Jaagsiekte sheep retrovirus (JSRV) which is the causative agent of a contagious lung cancer in sheep, called ovine pulmonary adenocarcinoma. Both share long signal peptides (96 and 84 amino acids, respectively) analogous to p14. Both accumulate in nucleoli where they colocalize with B23 (nucleophosmin). JSRV-SP also increases the nuclear export of unspliced viral RNA, comparable to p14 and HIV-Rev protein [81,82,83]. The three beta retroviral signal peptides share an unusually long tripartite structure, (similar to other signal peptides), with a hydrophilic, positively charged N-terminal (N) domain, a central hydrophobic domain (H) and a C-terminal (C) domain (Figure 4).

The phenomenon described above is not just a curiosity but rather suggests that due to their lengthy sequences, these signal peptides may express a broad spectrum of effects on the cellular physiology of retroviral infected cells, and thus are much more than just signal peptides.

## 6. Strategies for p14 Investigation in Human Breast Cancer

Once the nucleolar localization of p14 had been established, we embarked on additional (complementary) lines of p14 investigations, including:Diagnosis, of human breast cancer.Translational research involving the targeting, prevention and treatment of MMTV-associated murine lymphomas and mammary carcinomas.Assessing post-ER targeting functions of p14 and putative mechanisms involved.

### 6.1. p14 as a Diagnostic Marker for MMTV-Associated Breast Cancers

Collaborations with colleagues in the USA (NIH), Italy and Australia, where we contributed our unique anti-p14 polyclonal and monoclonal antibodies, resulted in the following (published) findings.

Formalin- fixed paraffin-embedded (FFPE) sections from 25 different human breast cancer samples on a commercial tissue array slide were subjected to immunohistochemical (IHC) analysis using a polyclonal anti-p14 antibody. Four sections (16%) were positive. Tissues adjacent to the positive tumors were negative. Duplicate sections were tested independently at the NIH with identical findings [55].

Feline and canine mammary carcinomas were analyzed for MMTV Env-like sequences by nested PCR, and for p14 by IHC. In the feline cases, 7% (6/86) were positive by both analyses, while in the canine samples, none were positive. Canine and feline normal mammary gland tissues scored negative by both PCR and IHC analyses [84].

IHC analysis of salivary gland tissue demonstrated clear positivity for p14 in cases positive by nested PCR, and the absence of p14 was documented in PCR-negative cases. Based on these and additional findings, human saliva was proposed as a route of inter-human infection for mouse mammary tumor virus [35].

A primary cell line, pBC, derived from a human invasive ductal breast carcinoma positive for MMTV Env-related sequences (through PCR) and p14 (through IHC), demonstrated nucleolar localization and cell surface expression characteristics of p14 [85]. Also, in lymph nodes infiltrated by the parental tumor, the tumor cells were p14-positive while the resident lymphocytes were negative.

To validate p14 as a target for diagnostic purposes, samples from a larger cohort of breast cancer patients were required. Indeed, a first attempt at this line of research was carried out by colleagues in Australia, using IHC with our anti-p14 antibodies [86]. As part of this study, 16 out of 42 women with breast cancer (38%) were positive for p14. Here too, lymph nodes infiltrated by the cancerous cells demonstrated positivity only for the cancer cells but not for the resident lymphocytes, attesting to the specificity of the analysis. In addition, 33 of these cases (both p14-positive and -negative) were also independently tested, in Italy, again using IHC with our anti-p14 antibodies. Here, 29/33 (≈90%) of the samples were identical, thus validating the Australian findings [86]. Interestingly, 7/13 women with benign hyperplasia, who subsequently (between 1–11 years) developed breast cancer, were also positive for p14 using IHC. The above findings demonstrate the significance of a reliable diagnostic test for MMTV-associated breast cancers at an early stage of tumor development. If positive, such findings could constitute a “game changer” in understanding and treatment of viral-associated breast cancer.

Figure 5 demonstrates the localization of p14 in nucleoli of murine and human cultured cells (Figure 5A–C) and IHC of a breast cancer tissue stained with polyclonal anti-p14 antibodies (Figure 5D–F) (adapted from Refs. [81,83]).

The above mentioned finding strongly suggest the validity and specificity of the antibodies applied.

### 6.2. Signal Peptide-Mediated Targeting, Prevention and Treatment of MMTV-Associated Murine Lymphoma and Mammary Carcinoma (Translational Research)

Cell surface expression of p14 (as target for immune surveillance): Flow cytometry of intact cells (a measure of cell surface expression) vs. permeabilized cells (whole cell expression), was carried out using polyclonal and monoclonal (M-66) anti-p14 antibodies. The cell lines tested were 4T1 and Mm5MT (murine mammary carcinomas) and T-25 (murine lymphoma). all expressing endogenous p14, as well as MCF-7 (a human breast cancer cell line), ectopically expressing p14, the pBC human primary breast cancer cell line shown to be positive for MMTV Env-like sequences, and p14. All revealed cell surface expression of p14 that ranged roughly between 5–10% of the entire cellular content of this signal peptide (Figure 6) [85].

Active immunization (preventive/protective vaccination): p14 is immunogenic. Inoculation of purified recombinant p14 into BALB/c mice induces the generation of serum anti-p14 antibodies as well as cytotoxic T-cells [85]. In addition, vaccination of mice with p14 (preventive vaccination) immunizes them against a subsequent challenge with murine lymphoma or mammary carcinoma cells that contain MMTV. Furthermore, adoptive T-cell transfer from p14-vaccinated mice into naïve mice immunized 100% of these hosts against a challenge with lymphoma cells that harbor MMTV. This was received at a ratio of 6/1 T-cells/lymphoma cells [85].

In vivo monoclonal antibody-mediated additivity and immune therapy: When mAbs M-66 and M-202 were injected separately (IP) into two groups of BALB/c mice previously inoculated with descendants of T-25 lymphoma cells, there was no documented difference in survival. None of the two groups survived longer than control mice inoculated solely with lymphoma cells. However, when mice challenged with 1.5 × 10^6^ lymphoma cells were injected with both mAbs at 1 mg/kg body weight every other day for 4 weeks, survival after 60 days was 87% as compared to 37% in the control (no antibodies) group [85], demonstrating clear efficacy as well as synergism in vivo between the two mAbs. Taken together, the monoclonal anti-p14 antibodies and adoptive T-cell transfer studies (passive immunization) demonstrate in vivo efficacy against aggressive mouse tumors (mammary carcinoma and lymphoma) that contain MMTV and express p14 on their surface [85].

### 6.3. Post ER Targeting Functions of p14

In addition to its role in ER targeting, nuclear export of unspliced viral RNA and immunogenic potential in active (vaccination) and passive (antibody and T-cell mediated) immune therapy, p14 was further analyzed for its cellular targets, mutations and transcriptional regulation.

Cellular protein targets: Nine p14 binding proteins were identified through independent purifications on two different p14 affinity columns, CNBr –based and Co^2+^ based (for His-tagged p14). Eluants were subjected to SDS-PAGE and the eluted proteins were sequenced. Four out of the nine identified p14 targets were nucleolar proteins [55]. Of particular interest were the nucleolar proteins B23 (Nucleophosmin) and the ribosomal protein L5 (RPL5), for which, in addition to ribosome biogenesis, shuttling functions, and a host of other functions, is also involved in stabilizing p53 levels in response to cellular stress. This is carried out through the inhibition of MDM2 activity, either via the 5SRNP–MDM2 pathway (L5) or through the ARF–MDM2 route (B23) (Figure 7). Thus, upon binding to L5 and B23, p14 may add a new level of hierarchy to the regulation of the cellular (nucleolar) stress response.

Impact of mutations: Based on sequence analysis, different mutations (point, deletion, stop codons) have been introduced along the sequence of p14 [87]. These were stably expressed in the MCF-7 human breast cancer cell line (devoid of MMTV) and used to probe into additional functions of p14. Of particular interest were two putative phosphorylation sites at serines 18 & 65. Analysis of point mutations in these sites, (Ser18Ala and Ser65Ala) demonstrated that p14 is a phospho-protein endogenously phosphorylated by two different serine kinases of the PKC and CK2 types, respectively [87].

Of the above mutants, those in CK2, PKC and CK2 + PKC (double mutant) phosphorylation sites demonstrated impaired, enhanced and impaired in vivo growth, respectively, upon inoculation into immune-deficient SCID mice when compared to wild type p14 construct. Thus, p14 (independent of the rest of MMTV) can function as either an oncogene (when phosphorylated by CK2) or as a tumor suppressor (when phosphorylated by PKC). The balance between the extent of phosphorylation of either site seems to determine the in vivo fate of MCF-7 cells bearing those mutants. Of additional interest are the findings that based on co-immunoprecipitation followed by Western blotting, the CK2 mutant (Ser65Ala) does not bind B23. Based on the same criteria, the PKC mutant (Ser18Ala) and wild-type p14 do bind B23. Deletion of amino acids 8–20 (named del1) also prevents B23 binding [87]. Thus, we conclude that the CK2 phosphorylation site (amino acids 65–68) and the del1 sequence minus amino acids 18–20 (PKC phosphorylation site) are both involved in the binding of p14 to B23.

The findings that p14 can express onco-protein function in vivo brings into light earlier reports, where an immune receptor tyrosine-based activation motif (ITAM) encoded within the env gene of MMTV was identified. Env-expressing human cells are transformed and become invasive in vitro [19,88]. Thus, it is intriguing to determine whether stable co-expression of ITAM and mutant(s) p14 in target cells that do not harbor either molecule will demonstrate any additivity or synergism in vitro or in vivo.

Transcription regulation: Using microarray analysis, we reported that p14 is also a transcriptional regulator of genes including L5 and ErbB4. There is up-regulation of these genes in its oncogenic (PKC mutant) configuration, and down regulation of these genes in its anti-oncogenic (CK2, and CK2-PKC double mutant) configurations, when compared to the wild type p14 construct [87].

## 7. Treatment Modalities Based on Target Localization

Currently, our lab has successfully applied p14 antibodies for diagnosis of MMTV-associated human breast cancer (as well as feline lymphoma ang mammary carcinoma), with colleagues in the USA, Italy and Australia [55,85,86,89,90,91]. In addition, p14 has been successfully used as a vaccine against an experimental lymphoma and mammary carcinomas that contain MMTV [85,92]. Monoclonal anti14 antibodies generated in our laboratory, as well as the use of adoptive T-cell transfer, impaired the in vivo growth of lymphoma and mammary carcinomas that harbor MMTV [85].

Based on the above findings, p14 seems to be a significant marker, both for diagnostic purposes as well as for various therapeutic scenarios in the clinical setting (Figure 8).

As stated above, p14 is located both intracellularly as well as on the surface of cells, tested so far, that harbor MMTV.

### 7.1. With Regards to Cell Surface Expression of p14

The idea that a cell surface marker due to MMTV could be found in human breast cancer would be a paradigm shift for the disease. Multiple uses can be considered in this respect: A—p14 as a therapeutic target using monoclonal humanized antibodies, single chain Fv and Bi-specific antibodies. Their mode of action has to be defined (ADCC, CDC, Apoptosis or other). B—Antibody-drug conjugates: these are effective cancer therapies where the cell surface marker is highly specific. Antibodies have been armed with radiolabels to deliver radiotherapy, and with very potent anticancer agents that cannot be given safely otherwise. For example, we have demonstrated that a conjugate between a monoclonal antibody directed against an intracellular epitope of the multi-drug transporter (ABCB1) and a translocator peptide (HIV-Tat), when incubated with multi-drug resistant cells, enters the cells, rendering them drug-sensitive [93,94]. C—A unique cell surface marker for MMTV could be useful in classifying breast cancer, and could lead to a new classification system, in which the MMTV-associated cancers are seen as a separate entity. D—A unique cell surface marker is useful in the diagnosis of breast cancer—sentinel node assessments, looking for early diagnosis of metastatic disease, are facilitated by antibodies recognizing tumor cells. Such a case where both the primary tumor and metastatic lymph node were positive for p14 was demonstrated in our studies [86]. E—Vaccination, for example, in women diagnosed early with benign hyperplasia positive for p14 [85,86], could be a very effective treatment modality. F—A unique marker could be used diagnostically—with a PET imaging label to localize the cancer. Such a marker could (if we can imagine that all the clinical validation were done) be used to assess the success of therapy—to show whether tumors were shrinking or not. It could also be used to identify patients whose disease was amenable to being targeted with one of the antibody conjugates. G—As p14 also induces a strong T-cell response, it is of obvious significance to look for anti-p14 TILs as well as generating CAR-T-cells with the scFv of the monoclonal anti-p14 antibody (with the highest affinity towards p14) as the recognizing component. These could also be used clinically in an effort to impair tumor growth and metastasis.

### 7.2. With Regards to Intracellular Expression of p14

A—Defining molecular mechanism(s) of p14 function: for example, the exact role of p14 in the regulation of the p53 checkpoint based on the abovementioned findings is necessary. B—Application of small molecule pharmaceuticals (inhibitors/enhancers) are of potential significance. For example, a small molecule that binds to p14 (with high affinity and specificity) and impairs its phosphorylation by CK2 might enhance the anti-oncogenic function of p14. The same outcome might be reached by using small molecule inhibitors of CK2 itself. These can be generated by various strategies, for example, by exposure of p14 to small molecule libraries or modeling based approaches. C—Determining whether an immune conjugate between a monoclonal anti-p14 antibody and a translocating peptide (i.e., HIV-Tat) can specifically enter target cells (expressing cell surface p14) and affect p14 function from inside as well as outside the cells will be important. D—α-Tubulin is another target of p14 identified through the two affinity columns and microsequencing approach [55]. The effects of its interaction with p14 and possible involvement in the anti-cancer therapeutics of anti-polymerization or anti-depolarization drugs combined with immune therapy has not yet been explored.

## 8. Concluding Remarks

While it remains unresolved whether MMTV/HBRV infection is an epiphenomenon or whether the virus plays a direct role in the pathogenesis of human cancer, our findings may indicate that the latter possibility is a valid one. If our and others’ findings regarding the involvement of MMTV in breast cancer are validated, then we have at our hands the armament needed for the early diagnosis, prevention and treatment of a significant proportion of breast cancers. Figure 8 describes our idea as to the road map to be taken for implementing the findings from “bench to bedside”.

## Figures and Tables

**Figure 1 viruses-14-02435-f001:**
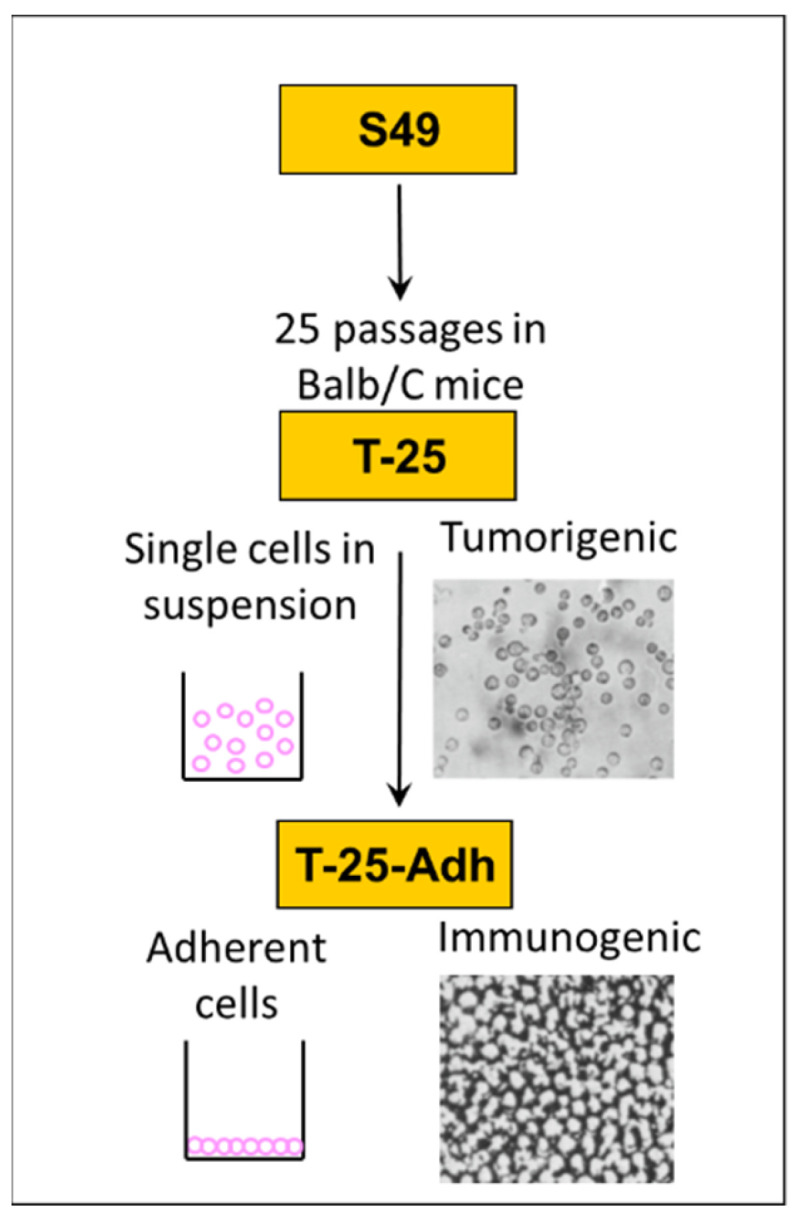
The S49 cell lineage. From highly tumorigenic T-25 cells that grow as single cells in suspension, non-tumorigenic (immunogenic) substrate adherent variants were derived through spontaneous selection.

**Figure 2 viruses-14-02435-f002:**
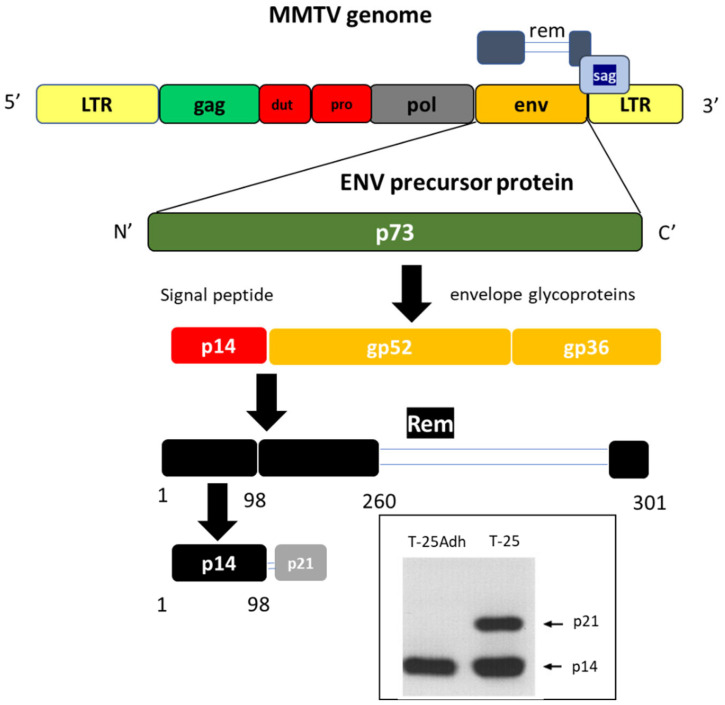
Protein products derived from the Env precursor protein of MMTV. p14 is the signal peptide. Rem and p21 are translation products of splice variants that include p14 with an extension into the envelope precursor sequence. Insert: Western blotting of T-25 and T-25-Adh cells with anti-p14. p21 is specific to T-25 cells and their direct tumorigenic descendants. (see text for details). Rem—Regulator of export/expression of MMTV, sag—Superantigen.

**Figure 3 viruses-14-02435-f003:**
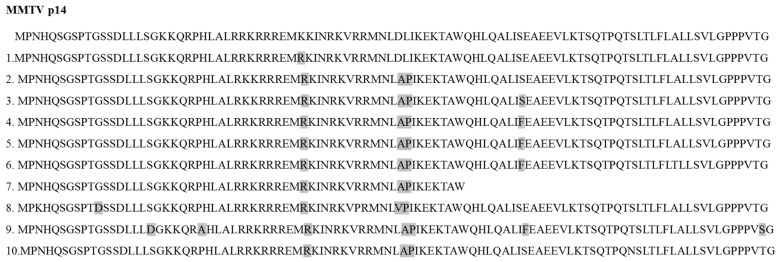
Alignment of p14 sequence (first line) with different mouse (MMTV 1–4) and human (HBRV 5–7) signal peptides taken from NCBI—BLASTp (No. 1–7) and a Signal Peptide Website (MMTV 8–10) (last updated by K. Kapp in 2010). Shaded amino acids demonstrate changes in the sequence between the different species as compared to MMTV-p14.

**Figure 4 viruses-14-02435-f004:**
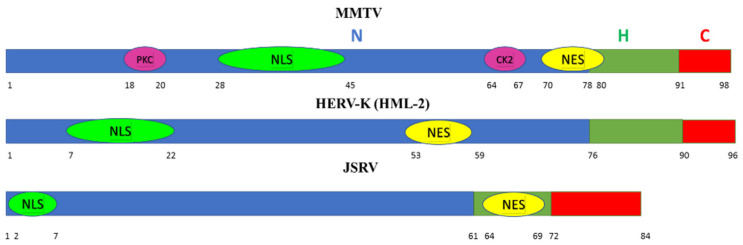
Tripartite composition of the three extended Betaretroviral signal peptides: MMTV, HERV-K (HML-2) and JSRV. Shown are relative positions of the different domains, N (Blue), H (Green) and C (Red), as well as selected positions of functional sequences; PKC and CK2 phosphorylation sites; NLS—Nuclear Localization Signal; NES—Nuclear Export Signal.

**Figure 5 viruses-14-02435-f005:**
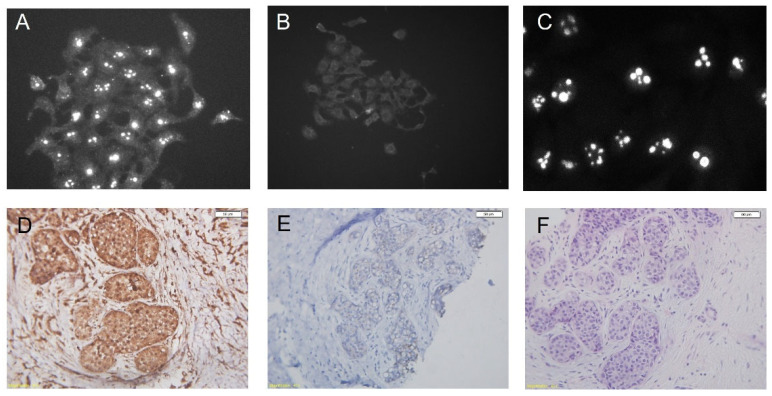
In vitro and b in vivo staining of p14. (**A**–**C**): Immune fluorescence (using Mab M202) of cultured MCF-7 cells that ectopically express p14, and parental MCF-7 cells that lack p14 and T-25 cells that contain p14, respectively. (**D**–**F**): Immunohistochemistry (peroxidase labeling) of a primary human breast tumor using polyclonal anti-p14 antibodies: (**D**)—Anti-p14 antibody; (**E**)—Control, no first antibody; (**F**)—H&E staining of the tumor. Bars—50 µm.

**Figure 6 viruses-14-02435-f006:**
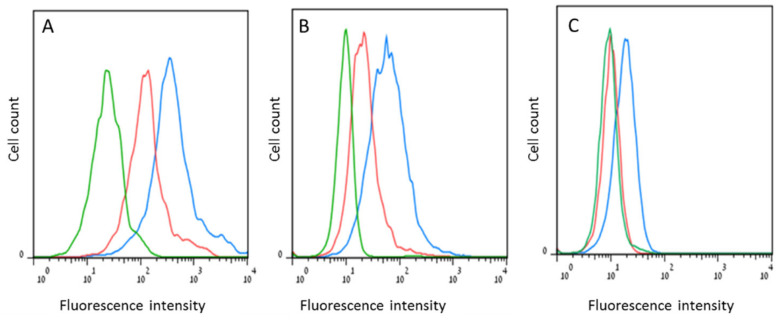
Cell surface vs. whole cell expression of p14 using flow cytometry and Mab M-66 as first antibodies. (**A**) T-25 cells that contain MMTV; (**B**) MCF-7 cells ectopically expressing p14: (**C**) MCF-7 cells devoid of p14. Green—Control, no. 1st antibody; Red—Intact cells (cell surface expression; Blue—Permeabilized cells (whole cell expression). Adapted from Ref. [81].

**Figure 7 viruses-14-02435-f007:**
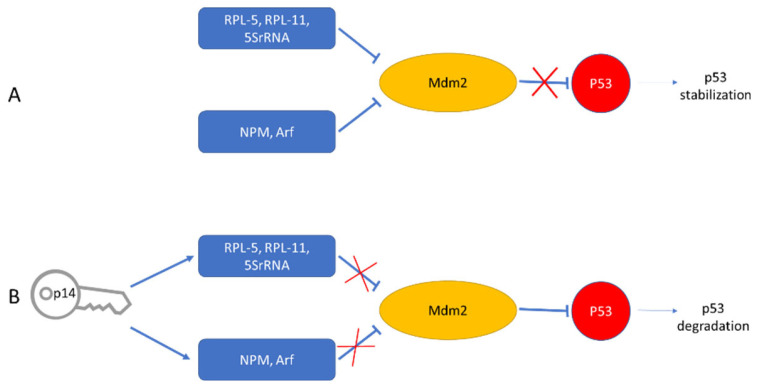
Model illustrating a proposed effect of p14 on p53 checkpoint. (**A**) In the absence of p14, MDM2 is inactivated by the NPM (B23) and RPL5 complexes, thus stabilizing p53. (**B**) In the presence of p14, the two complexes become inactive. In this case, p53 is degraded.

**Figure 8 viruses-14-02435-f008:**
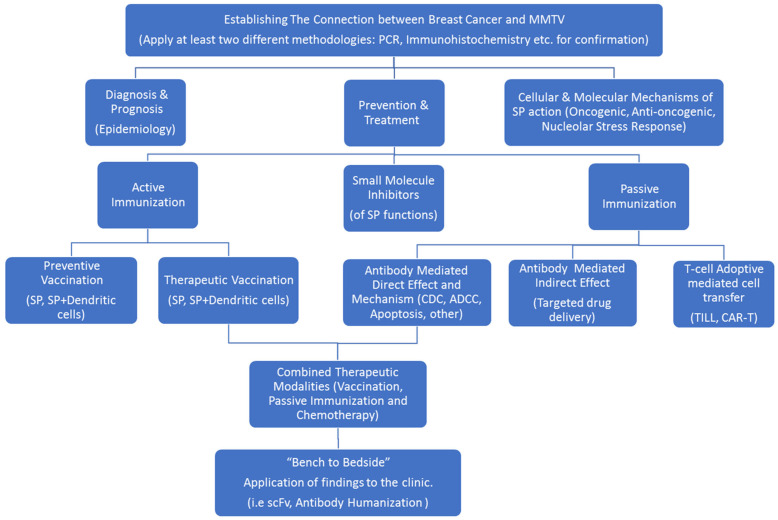
Signal-peptide-based road map for the diagnosis, prevention and treatment of MMTV (HMTV, HBRV)-associated breast cancer and other maladies (i.e., primary biliary cholangitis) associated with the virus in the clinical setting.

## Data Availability

Not applicable.

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
