# Peer review of "Life after Cleavage: The Story of a β-Retroviral (MMTV) Signal Peptide—From Murine Lymphoma to Human Breast Cancer"

_viruses, 2022, doi:10.3390/v14112435_

Round 1

Reviewer 1 Report (New Reviewer)

Manuscript ID

viruses-1933029

Title: Life after Cleavage: The Story of a β-retroviral (MMTV) Signal Peptide - From Murine Lymphoma to Human Breast Cancer

Authors:  Jacob Hochman and Ori Braitbard

Review

This is a long over-due review by an expert that makes important contributions to the understanding of betaretrovirus biology and pathogenesis.  Specifically, it discusses the history and role of a unique viral protein, the p14 signal peptide, arising from the Env and Rem open reading frames (ORF) of the mouse mammary tumor virus (MMTV), a virus that causes breast cancer and leukemia/lymphomas in mice.  Importantly, MMTV is increasingly being implicated in the etiology of human breast cancer and other diseases as well, such as lymphomas and primary biliary cirrhosis. 

Overall, the review is quite extensive and covers most areas related to p14.  However, there are some areas of concerns that should be addressed to improve the manuscript before it can be published:

1.     The figures are quite basic and can be improved.  Figures 1, 2, 4, & 7 are too big and can be reduced in size. 

2.     The MMTV genome in Fig. 2 is incorrect and does not show dUTPase, Rem or Sag ORF.  It should be corrected and overall the figure refined.

3.     Resolution scale should be mentioned in the microscopic images in Fig. 5.

4.     Since this review is about p14, the part on its biogenesis, mechanism of retrotranslocation, as well as unconventional ER trafficking are missing.  A paragraph or two covering these aspects should be added to the review to make it more comprehensive after line 117 along with related references that should be added, such as:

a.     Byun et al. Retroviral Rem protein requires processing by s,ignal peptidase and retrotranslocation for nuclear function. Proc Natl Acad Sci U S A. 2010 Jul 6;107(27):12287-92. doi: 10.1073/pnas.1004303107.

b.     Byun et al., Requirements for mouse mammary tumor virus Rem signal peptide processing and function. J Virol. 2012 Jan;86(1):214-25. doi: 10.1128/JVI.06197-11.

c.     Byun et al., Mouse Mammary Tumor Virus Signal Peptide Uses a Novel p97-Dependent and Derlin-Independent Retrotranslocation Mechanism to Escape Proteasomal Degradation. mBio. 2017 Mar 28;8(2):e00328-17. doi: 10.1128/mBio.00328-17.

d.     Xu et al., Unconventional p97/VCP-Mediated Endoplasmic Reticulum-to-Endosome Trafficking of a Retroviral Protein. J Virol. 2021 Jun 24;95(14):e0053121. doi: 10.1128/JVI.00531-21.

e.     Das P, et al., A Retrotranslocation Assay That Predicts Defective VCP/p97-Mediated Trafficking of a Retroviral Signal Peptide. mBio. 2022 Jan 4;13(1):e0295321. doi: 10.1128/mBio.02953-21.

5.     The manuscript requires editing for English language and clarity. A marked manuscript file with edits is being provided that should be used by the authors for making corrections.  Additionally, the overall language of the manuscript needs revisions throughout the since at many places, it is anecdotal.

6.       The “Concluding remarks and future prospects” section which is currently no 7 should be the last section (no 8) of the manuscript, while the current section 8 on “Treatment modalities based on target localization” should be section 7.  Concluding remarks could start with line 377.

Author Response

see the attachment

Reviewer 2 Report (New Reviewer)

Review of Life after Cleavage: The Story of a β-retroviral (MMTV) Signal Peptide - From Murine Lymphoma to Human Breast Cancer by Hochman et al in Viruses

 The authors provide a review of a very interesting topic of MMTV and its link to human breast cancer. The interest in the topic is apparent with over 70 reviews on MMTV and breast cancer. However, a Pubmed search for reviews on MMTV p14 and cancer is surprisingly lacking making this review needed. It is a well written review that is easy to read and follow.

The reviewer appreciated the balanced approach of the introduction to if there is a direct link between MMTV and breast cancer but would like to see this paragraph expanded in a quick synopsis of the opposing literature cited.

“However, there is an accumulation of reports that question the involvement of MMTV in human breast cancer. Some of these are presented and discussed in the following references22,23,2425.”

Overall the suggestions are minimal and the reviewer recommends acceptance if the following is addressed.

 Major points

1-    It is not clear if figures 5 and 6 are new data using the methods in the references or old data adapted from publications 60 and 62… This should be clarified.

2-    While not the point of the review a quick discussion of MMTV lifecycle, replication, and the role of Env could be helpful for the general reader coming from the cancer angle.

Minor points

1-    It would be nice if the gene products of MMTV genome (Fig 2) were drawn to proportion to their sizes like the rest of the figure.

2-    Fig 3 – can this figure be adapted so that the sequences are not separated by the titles… ie it would be easier to see.

Round 2

Reviewer 1 Report (New Reviewer)

The authors have made a good effort to respond to this reviewer.  The manuscript is reading much better, but there still are some incorrect statements, technical, and editorial issues which need to be addressed before the article can be published.  

Lines 30-37:  This is this the first section in the review which begins by the heading “MMTV and Cancer”, yet there is no mention of MMTV in these lines!   This can be fixed by keeping lines 38-43 from the original submission after line 37 and then adding a statement (highlighted in yellow below) before starting Section 1.1 on the MMTV genome.  This is shown below.  This means that the authors will need to remove lines 71-74 from the revised submission.

1. MMTV and Cancer

It is estimated that 15-20% of all cancers are associated with viruses. Viruses known

to cause cancer in humans include the Epstein-Barr virus (EBV), hepatitis B virus (HBV),

human T-cell lymphotropic virus type 1 (HTLV-1), human papilloma virus (HPV), hepa-

titis C virus (HCV), Kaposi's sarcoma-associated herpes virus (KSHV) and the Merkel cell

polyomavirus (MCV)1. The increasing interest and better understanding of the role of

viruses in cancer pathogenesis led to the development of preventive vaccines against HBV

and HPV, as well as directly contributed to fundamental discoveries in cancer biology.

Mouse Mammary Tumor Virus (MMTV) is a member of the genus Betaretroviruses.

It causes mammary carcinoma (breast cancer) and lymphoma in mice2,3,4,5. Since the ad-

vent of PCR technology, an increasing body of evidence supported the involvement of a

virus, that is genetically indistinguishable from MMTV (also named HMTV for Human

mammary tumor virus or HBRV for Human Betaretrovirus), anywhere from 16%-38% of

human sporadic breast cancer6,7,8,9.  However, this is a highly controversial area that we would like to address by using evidence based on the association of the p14 signal peptide of MMTV with human breast cancer.

Line 40: Although it is true that MMTV env is non-overlapping with the gag/pro/pol genes, but rem and sag genes overlap with the env reading frame.  Therefore, env cannot be called a “non-overlapping” gene and it is better to remove the word “non-overlapping” from the text.

Line 57-58.  Similarly, while it is true that MMTV interacts with its receptor (TfR1) at the cell surface, the virus is actually endocytosed along with the TfR1 to a late endosomal acidic compartment where the actual fusion of the virus takes place, not on the cell surface like other retroviruses.  It is at this point that the capsid is released into the cytoplasm for further uncoating (reviewed in Ross, 2010).  Therefore, please make the correction and add the reference.

Line 66-68:  Please modify this part as follows:

While little is known about where and how MMTV virion assembly takes place, there is a lot more clarity on how the MMTV genomic RNA gets packaged into the virus particle using structural RNA elements on the genomic RNA (Mustafa et al., 2012: https://doi.org/10.1371/journal.pone.0047088; Akhtar et al., 2014: https://doi.org/10.1186/s12977-014-0096-6; Mustafa et al., 2018: https://doi.org/10.1080/15476286.2018.1486661: Chameettachal et al., 2021: https://doi.org/10.1093/nar/gkab223)

Line 90:  Remove “s” from questions.

Line 92:  Remove the period after the references.

Line 116-117:  … (that harbor MMTV) and grow ….

Line 149: Superantigen is one word.

Line 161.  This line should start a new paragraph, but it looks like it is continuing a previous sentence.  Please check and remove “60.” from the start of the sentence. 

Line 161.  Remove “a” before proposed.

Lines 365-372.  Somehow, the reviewer missed that this section is not supported by any references, despite making big claims!  This section needs to be supported by published references.  Otherwise, it needs to be reworded so that it is clear that this work is in progress and if proven true should back the claims of the involvement of MMTV in human diseases and how therapeutics based on p14 could be used for treatment modalities.

Author Response

Comment related to lines 30-37: Paragraphs have been edited as suggested.
Comment related to line 40: Done.
Comment related to lines 57-58: Correction was made and references (including the Ross review) were added.
Comment related to lines 66-68:Modified plus the suggested references.
Comments related to lines 90, 92, 116-117,149,161:Done.
Comment related to lines 365-372: References of published work to support the claims have been added (both our own publications as well as publications that have used our anti-p14 antibodies).
We trust that the manuscript is now accepted for publication by the issue of VIRUSES edited by Prof. A. Mason

This manuscript is a resubmission of an earlier submission. The following is a list of the peer review reports and author responses from that submission.

Round 1

Reviewer 1 Report

The paper is a review of the previous work conducted by the authors on protein p14 and related findings. Data are highly relevant, as demonstrated by the quality of the journals on which their results were published: Nature, J Cell Biol, Cancer Immunol Immunother, J Natl Cancer Inst, Virology etc.

In 1983 Zotter S and colleague published a paper in which a MMTV p14 protein is mentioned. Could authors comment these previous findings specifying the eventual relationships between that p14 and their p14?

Zotter S, Grossmann H, François C, Kozma S, Hainaut P, Calberg-Bacq CM, Osterrieth PM. Among the human antibodies reacting with intracytoplasmic a particles of mouse mammary tumor virus, some react with MMTV p14, the nucleic-acid-binding protein, and others with MMTV p28, the main core protein. Int J Cancer. 1983 Jul 15;32(1):27-35. doi: 10.1002/ijc.2910320106.

The last section of the paper “Treatment modalities based on target localization” is excessively long and too general, with hypothetical uses that do not have (or do not have enough) scientific support. Maybe Authors could reduce the paragraph to a short comment of Fig 10, focusing only on vaccines.

The paper needs a careful editing: many unnecessary capital letters, too many letters and numbers. For instance, paragraph 5. P14 as diagnostic marker…, maybe it is better to have just a dash, without 1A, 1B etc. Even because at the end of the paragraph there is a 1) Signal peptide targeting, prevention …. followed by 2A-… and then 2B etc. Later there is 2). p14, as a multi-functional signal peptide….. followed by 3A, 3B etc.

The effect is confusing and reading becomes difficult.

The paper is interesting and deserves to be published on the issue dedicated to HBRV after minor revision.

Author Response

The manuscript has been edited accordingly. 

Reviewer 2 Report

This is an interesting paper that summarizes what is known about the MMTV p14 protein and as such it should be of interest to the readership of “Viruses”. However, some modifications should be made before this paper can be published, including

  • the authors must cite Redmond and Dickson, 1983 https://pubmed.ncbi.nlm.nih.gov/11894899/ on page 2, at the end of the sentence “synthesized as a 73-kDa Env-precursor. Its signal peptide (p14) is cleaved and the protein is further processed to give the envelope glycoproteins, gp52 and gp36 (Fig.2).” Indeed, these authors were the first to show that the MMTV precursor is a long peptide and to postulate a role for this in the virus life cycle and this should be commented on.
  • Since this is a review, the authors should consider discussing the second promoter located in the 5’ LTR as detected by the Gunzburg group (https://pubmed.ncbi.nlm.nih.gov/8391646 and https://pubmed.ncbi.nlm.nih.gov/10684308/) and the intriguing possibility that is could be used for transcription of p14
  • Also the authors should consider discussing the potential role of other MMTV encoded factors that may synergise in tumorigenicity such as the ITAM motif in the envelope gene as shown by the group of Ross https://pubmed.ncbi.nlm.nih.gov/15684322/
  • The use of the numbering from page 5 onwards (1A-1E, 2A-2E, 3A-3C) is confusing and this system should not be used. 

Minor points

  • There are numerous citations where the reference number is not given superscript including (but not limited to):-

Page 1 “polyomavirus (MCV)1” should be “polyomavirus (MCV)1”

Page 1, line 44 “similar to the mouse virus10” should be “similar to the mouse virus10

Page 3 superscript needed for references in the following text “cell evasion pathways49. In another report50”

The paper needs to be checked for such occurrences and corrected.

  • Many words are given starting with a capital letter including (but not limited to):-

Page 2, Line 50 “Also, Saliva has been..” should be “Also, saliva has been…”

Page 2, line 55 “mice supporting Zoonosis” should be “mice supporting zoonosis”

Page 5, line 152 “..associated with Germ Cell tumors and Melanoma, and the…” should be “associated with germ cell tumors and melanoma, and the Jaagsiekte Sheep…”

The paper needs to be checked for such occurrences and corrected.

  • Page 1 “a virus,” should be “a virus” (no comma)

Author Response

major comments:

  1. The manuscript by Redmond and Dixon has been addressed and cited.
  2. As the present manuscript deals with MMTV solely at the protein level and not at the DNA regulatory level, we think that discussing different promoters, while of interest, is not instrumental to this particular review.
  3. Point well taken. A paragraph that draws attention to ITAM as putative synergistic element to p14 has been added (see lines 337 - 342).
  4. This has been taken care of in accordance with the comments (numbering deleted).

Minor comments: These have been corrected as suggested by the reviewer.

Reviewer 3 Report

This review article by Hochman and Braitbard focuses on the role of the signal peptide (SP) encoded by mouse mammary tumor virus (MMTV).  The manuscript focuses heavily on the author’s work and introduces new data, which should be reviewed separately.  Previous work by this author suggests that the SP, called p14 in the author’s papers, is present on the surface of MMTV-infected cells, although the major portion of this protein is localized to the nucleolus. The primary role of MMTV-encoded SP has been shown to be similar to that of HIV-1 Rev in mRNA export and expression.  The manuscript argues for the utility of p14 as a marker for human breast cancer, although a role for MMTV in this disease has not been demonstrated. A review article on the role of p14, particularly related to cancer, is not warranted.

Major comments:

  1. The article reads like a primary publication with a long introduction.
  2. The review of the literature only presents the positive association between MMTV and human breast cancer or biliary cirrhosis. The more mainstream view that MMTV is a passenger virus in these studies has not been given.
  3. Even though it has been proposed, saliva is not a major route of transmission for animal retroviruses (line 50). Blood, sex, and milk-borne transmission are the accepted methods. 
  4. 5 - It is not clear that the S49 lymphoma cells are producing infectious MMTV or that they express functional p14. Human breast cancer cells expressing p14 appear to be transfected rather than infected with MMTV.  If human tumors don’t express p14, the protein is not useful as a tumor marker or a target for immunotherapy.
  5. 6-8 appear to be primary data that have not been reviewed. Their characterization does not belong in a review article.
  6. 9 is not supported by experimental results reviewed here.

Minor comments:

  1. Figure legends are insufficiently descriptive.
  2. The Genus name is Betaretrovirus (not beta-retrovirus).
  3. 2 – the diagram of the MMTV genome is not accurate.
  4. 3 – the alignment of sequences is not useful if the sequences are not of human origin.
  5. Line 226 – Do the authors mean that the surface expression of p14 is 5-10% of the total p14 in the cells?

Author Response

major comments:

1 and 5. The manuscript presents the extensive research carried out in our laboratory in collaboration with colleagues in Australia, Italy and the USA, in an effort to better understand the involvement of p14 (as part of MMTV) in murine and human cancer. Most of the results presented have been published in scientific journals and meetings, or taken from available data bases. Results that have not been published: a) Mainly figure 8 that demonstrates possibilities for in vitro synergy between different p14 Mabs when the (most important in vivo synergy has already been established and published in ref. 55). b) Epitope mapping of p14 has already been published for Mab M-66 (ref. 31) and just extended here using the exact approach for 2 additional Mabs. This is also related to in the manuscript itself. c) Figure 6 presents data already published in a different format (ref.60). References to the original studies are added to the legend to figures where appropriate.  

  1. We have added 4 references that question the involvement of MMTV in breast cancer (refs. 22-25).
  2. Saliva has only been suggested as an option for inter-human transmission of the virus.
  3. It was demonstrated that 16/42 human breast cancer cases expressed p14 (see ref. 61) and as such, p14 can serve as a diagnostic marker for these cases, independent of whether it fulfills (or not) any function. These cases have definitely not been transfected with MMTV. Also, breast cancer cells that demonstrated MMTV Env like sequences, and a primary cell line derived from them, were positive for p14 (again, serving at least as diagnostic marker).
  4. See above.
  5. We have slightly changed the legend to Fig.9 that stresses  the fact that at this point in time the proposed hierarchy is of a speculative nature.

Minor comments:

  1. Where applicable we extended the legend to figures

2 .Point well taken

  1. We have slightly modified the diagram. The Envelope DNA is transcribed and translated to the p73 envelope precursor protein which is subsequently cleaved to produce Rem, p14 and p21.This sentence has been added to the legend of figure 2.
  2. A Major point of the manuscript is the road taken from murine lymphoma to human breast cancer (see title). It is thus essential that both mouse as well as human sequences containing p14 are compared. Indeed, we have added additional mouse and human sequences that have been deposited in the NCBI-BLASTP (see Fig.3).  
  3. Yes!

Round 2

Reviewer 3 Report

The revised manuscript by Hochman and Braitbard has provided minimal changes to the original manuscript.  The authors accept the idea that MMTV is the cause of human cancers when compelling evidence has not been provided in this article or elsewhere. This review is not of broad significance to the field because the premise that MMTV signal peptide is a reasonable therapeutic target for human breast cancers has not been validated.